# Factors associated with unsuccessful tuberculosis treatment among homeless persons in Brazil: A retrospective cohort study from 2015 to 2020

**Osiyallê Akanni Silva Rodrigues**[1]*, **Hammed Oladeji Mogaji**[1], **Layana Costa Alves**[1], **Renzo Flores-Ortiz**[2], **Cleber Cremonese**[1], **Joilda Silva Nery**[1]

1 Institute of Collective Health, Federal University of Bahia, Salvador, Brazil, 2 Centre for Data and Knowledge Integration for Health, Oswaldo Cruz Foundation, Salvador, Brazil

* osi.nutri@gmail.com

**Data Availability Statement:** The datasets used and/or analyzed during the current study are extremely large and available at https://zenodo.org/record/8001049. The analysis code has also been

## Abstract

### Background

Tuberculosis (TB) is a preventable and a curable disease. In Brazil, TB treatment outcomes are particularly worse among homeless populations who are either of black race, malnourished or living with HIV/AIDS and other comorbidities. This study therefore evaluated factors associated with unsuccessful TB treatment among homeless population (HP) compared to those with shelter.

### Methodology/Principal findings

The study population was composed of 284,874 people diagnosed with TB in Brazil between 2015 and 2020 and reported in the Information System for Notifiable Diseases (SINAN), among which 7,749 (2.72%) were homeless and 277,125 (97.28%) were sheltered. Cox regression analysis was performed with both populations to identify factors associated with unsuccessful TB treatment, and significant predictors of TB treatment outcomes. Results show that HP are more susceptible to unfavorable outcomes when compared to sheltered people (Hazard Ratio (HR): 2.04, 95% CI 1.82–2.28). Among the HP, illicit drug use (HR: 1.38, 95% CI 1.09–1.74), mental disorders (HR: 2.12, 95% CI 1.08–4.15) and not receiving directed observed treatment (DOT) (HR: 18.37, 95% CI 12.23–27.58) are significant predictors of poor treatment outcomes. The use of illicit drugs (HR: 1.53, 95% CI 1.21–1.93) and lack of DOT (HR: 17.97, 95% CI 11.71–27.59) are associated with loss to follow-up, while lack of DOT (HR: 15.66, 95% CI 4.79–51.15) was associated with mortality among TB patients.

### Conclusion/Significance

Homeless population living in Brazil are twice at risk of having an unsuccessful treatment, compared to those who are sheltered, with illicit drugs use, mental disorders and lack of DOT as risk factors for unsuccessful TB outcomes. Our findings reinforce the arguments for

provided as supplementary file. (S1 File Data frame).

**Funding:** The author(s) received no specific funding for this work.

**Competing interests:** The authors have declared that no competing interests exist.

an intersectoral and integral approach to address these determinants of health among the vulnerable homeless populations.

## Author summary

Tuberculosis is one of the world's most deadly disease and is responsible for 4,000 deaths daily. People living in impoverished conditions, especially those who are homeless are among the worst affected. In Brazil, the population of homeless persons is rising. This population is affected by socioeconomic issues such as worst jobs, structural racism, lack of civil rights, limited access to health care service and stigmatization. Our study therefore investigated the reasons why tuberculosis treatments are unsuccessful among homeless people in Brazil in comparison to populations with shelter. We found out that homeless population living in Brazil are twice at risk of having an unsuccessful treatment, compared to those who are with shelter. In addition, we found that homeless population who use illicit drugs and had mental illness were the most affected. Also, the non-implementation of directly observed treatment is a major reason why most patients do not complete their treatment, and a major predictor of death among these homeless populations. Since tuberculosis is a preventable and curable disease and diagnosis and treatment are freely available in the Unified Health System (SUS, *Sistema Único de Saúde*) of Brazil, it therefore becomes important to understand and address these barriers that have limited the access of homeless populations to existing health services.

## Introduction

Tuberculosis (TB) is a preventable and curable communicable disease and remains one of the most important infectious diseases in Brazil, hence the Unified Health System (SUS, *Sistema Único de Saúde*) was established to provide free diagnosis and treatment services to populations in Brazil [1]. Before the advent of COVID-19 pandemic, TB was responsible for most deaths attributed to infectious diseases [2]. In Brazil, TB distribution is associated with socioeconomic realities, with higher incidences recorded in places with worst socioeconomic scenarios such as malnutrition, poor housing and working conditions [3,4]. Furthermore, these poor socioeconomic contexts also make it difficult for populations to access health care services for TB [3,4]. Among vulnerable groups, the homeless population (HP) remains heavily exposed with higher morbidity and mortality chances compared to vulnerable populations with homes [5,6,7].

Over the last decades, the number of HP have risen significantly. This social phenomenon started during pre-industrial period in Europe due to the ambitious capitalist production and, consequently, loss of peasant lands [8]. In Brazil, HP faces poverty, social exclusion, difficulties to access health services, prejudice, stigma, and mostly utilize public and degraded area for housing and livelihood [9,10]. Most of them become homeless because of addiction to excessive alcohol consumption and drug abuse, unemployment and family disagreement [8]. According to the 2021 Unified Registry for Social Programs (*CADUNICO, Cadastro Único para Programas Sociais)*, there were 160,097 HP that received some social benefit in Brazil, with 88.8% classified to live in extreme poverty condition and 67.8% of them are of black or mixed ethnicity [11].

Numerous studies have investigated the higher risk of TB and comorbidities among HP compared to the general population, including studies conducted in Brazil and other countries

[5,7,12,13]. However, recent research focusing on the epidemiology of TB among this population has predominantly used a cross-sectional approach to estimate risk [7]. As a result, there is a lack of longitudinal studies that specifically examine the TB epidemiology among high-risk populations. Longitudinal studies are essential as they allow for the analysis of trends and patterns over time, offering a more comprehensive understanding of the factors associated with TB in this group. Such studies can provide valuable insights to support ongoing healthcare interventions targeted at high-risk populations. This study therefore employed a longitudinal approach in assessing the social and health determinants that influences TB treatment and outcomes among HP in Brazil. This study therefore evaluated factors associated with unsuccessful TB treatment outcomes among HP compared to those with shelter, with three sub-objectives; (1) explore the relationship between homelessness and TB treatment outcome, (2) identify factors associated with unsuccessful TB treatment among homeless and sheltered populations, and (3) determine the factors associated with undesirable treatment outcomes among homeless populations.

## Materials and methods

### Study design and population

This is a retrospective cohort study of people, homeless or sheltered, diagnosed with TB in Brazil from 2015 to 2020.

### Data source and collection

De-identified records from the Information System for Notifiable Diseases *(Sistema de Informação de Agravos de Notificação*, SINAN). The SINAN was established in 1993 and has advanced to become a robust data integration platform for the health system at the municipal, state and federal government levels [14]. The system is fed with notifications of cases with compulsory notification from different agencies across the country, to support the planning and evaluation of health interventions and other health priorities [14,15]. As such, TB confirmed cases and reports on follow-up are registered electronically by health-care workers across different agencies and subsequently uploaded to Ministry of Health and SINAN [16,17]. In Brazil, TB treatment is implemented according to the guidelines established by the Brazilian Ministry of Health. These treatment services are accessible to all citizens through the public healthcare system known as *Sistema Único de Saúde* (SUS). The treatment is provided free of charge and follows a daily-dosage regimen of Directly Observed Therapy (DOT) over a six-month period. However, there are variations in the treatment models offered across clinics, with some providing self-administered therapy (SAT). The frequency of supervision during the treatment may also vary, with at least three visits per week in the first two months and twice weekly visits for the remaining four months [18]. For this study, anonymized reports on TB were obtained from SINAN following an approved request to the Federal Ministry of Health.

### Inclusion and exclusion criteria

The inclusion criteria for this study were patients with: (1) new cases of TB, and (2) between age 18 to 90 years. Exclusion criteria includes: (1) TB patients without a clear definition of homelessness or been sheltered, (2) Patients with unknown previous TB treatment history (3) previously treated patients with relapse, (4) patients who were transferred, (5) patients treated after loss to follow-up, (6) post-death cases, (7) patients with drug resistant TB (DR-TB), (8) patients with primary abandonment, and (9) patients who changed therapy.

## Outcome variable and period of the follow-up

A dichotomous dependent variable was generated (TB outcome) with two categories: unsuccessful and successful TB treatment. Based on the literature [19–21], we defined unsuccessful treatment as falling into one of the following categories: (1) treatment failure, (2) death, (3) loss to follow-up, or (4) not evaluated. The "Treatment failure" category refers to patients who tested positive for sputum smear or culture at the 5th month of treatment or later. The "Death" category includes patients who died due to TB or other causes. The "Loss to follow-up" category pertains to TB patients who had taken medications for 30 days or more but then interrupted treatment for one month or longer. The "Not evaluated" category describes cases where TB patients were either transferred to other healthcare units or their records are unknown, incomplete, or have missing information. On the other hand, the successful treatment category refers to patients who are considered cured if they have two negative smears, one during any month of treatment and the other at the 5th or 6th month of therapy, or if they completed treatment without any signs of failure and were discharged based on clinical and radiological examinations [1,2,19].

The follow-up period was determined by calculating the difference between the start and end dates of TB treatment. Although the standard treatment for TB is recommended to be 180 days [22], there are often delays in recording and reporting treatment completion in the national system (SINAN). These delays can affect the closure dates of cases in the SINAN system. Additionally, SINAN allows a deadline of nine months or more for opening a notified case and concluding treatment [11]. To ensure accurate reporting, we extended the study period to 270 days. Therefore, we included TB treatment notifications recorded from January 1, 2015, until December 31, 2020, with therapy closure within this extended period. We performed descriptive sensitivity analyses by comparing treatment periods of 180, 270, and 360 days to evaluate their impact on the outcome results. The findings revealed a p-value below 0.05, indicating that the observed differences are statistically significant and not merely due to chance (see S1 and S2 Tables for details).

## Covariates

The covariates included in this study were selected according to the vulnerability index in the conceptual framework for determinants of TB in Brazil [23] and other sociodemographic and clinical characteristics of the patients (S1 Fig). Sociodemographic covariates include sex (female or male), race (white, black or mixed ethnicity, Asian and indigenous), age groups (18–29, 30–39, 40–49, 50–59, 60–69 and 70–90), level of education (bachelor and more, tertiary education, secondary education or illiterate), beneficiary of government cash transfer program (yes or no), and region of Brazil (Southeast, North, Northeast, Central-west or South). However, clinical covariates include co-infection with HIV (yes or no), alcohol use (yes or no), diabetes (yes or no), mental disorder (yes or no), tobacco use (yes or no), illicit drug use (yes or no) and directly observed treatment (DOT) (yes or no). Clinical forms, pulmonary or extrapulmonary, were also included. Patients with both pulmonary and extrapulmonary presentation were classified as "pulmonary" [24,25].

## Statistical analysis

The process of statistical analysis was broken down into four distinct stages, which include (1) descriptive analysis to generate absolute and relative frequencies, (2) Cox regression analysis between homeless and sheltered populations to compare risk and outcomes, (3) Cox model to identify factors associated with unsuccessful treatment among HP, and (4) Cox model to identify the magnitude of association between significant predictor and each outcome category.

For the descriptive analysis, missing values were considered, and associations were measured using Chi-square test for independence or Fisher's exact test for expected frequencies lower than 5 [26]. Probability values that are equal or less than 0.05 were considered statically significant. However, the survival analysis was performed only with complete entries. As such, we dropped records with missing values from the variables "level of education", "DOT", "HIV", and "race". Following that, for the Cox regression, we removed the covariates "region of Brazil" and "beneficiary of government cash transfer program" as they presented incomplete categories after filtering the database. Initially the crude and full regression were conducted to compare the risk of unsuccessful treatment among both homeless and sheltered populations with influence of the covariates. The crude and adjusted Cox hazards regression (HR) was also performed to identify the factors associated with the unsuccessful treatment among the HP and, also the association between significant predictors (illicit drug use, mental disorder and DOTs) and TB outcome categories. Results were significant when Wald test p-values is less than 0.05 at 95% confidence interval. A cumulative incidence curve was used to display the unsuccessful outcome in both groups along the period of follow-up, including all covariates. Data management and visualizations were carried out in RStudio version 4.2.1. The analysis code has been provided as supplementary file (S1 File Analysis Code).

## Ethics

The study protocol was submitted to the unified Brazilian platform for registration of research involving human beings (*Plataforma Brasil*) and approved by the Ethics Committee of the Institute of Collective Health of the Federal University of Bahia in December 2021 (Registration number 53542621.7.0000.5030).

## Results

### Overview of the dataset

A total of 284,874 individuals were registered as new TB case across the study years, comprising of HP (n = 7,749; 2.72%) and those who are sheltered (n = 277,125; 97.28%). To perform the survival analysis with complete records, 252,212 missing values were removed. The entire flowchart of the analysis procedures is presented as S2 Fig.

### Sociodemographic and clinical characteristics of the study population

Table 1 shows that majority of HP diagnosed with TB were men (83.21%, n = 6,448), of black or mixed ethnicity (59.14%, n = 4,583), aged between 30 to 39 years (31.50%, n = 2,441), and had secondary education (11.43%, n = 886). Additionally, most of the HP with TB are users of alcohol (54.27%, n = 4,205), tobacco (47.24%, n = 3,661) and illicit drugs (53.03%, n = 4,109). After 270 days of treatment, 75.47% of population with homes were successfully treated of TB, in contrast to 39.84% of HP. The proportion of HP who were loss to follow-up is three times higher than those in the sheltered group (36.02% *vs* 10.81%). S3 Fig. shows the cumulative incidence for TB were higher over the study period among HP who had unsuccessful TB treatments compared to the sheltered populations. Also, death was more common among the HP compared to those in the sheltered group (14.37% *vs* 8.24%). (Table 1).

### Association between sociodemographic and clinical covariates and the risk of unsuccessful TB treatments among the study populations in Brazil (2015–2020)

Table 2 shows that the HP are more likely to have an unsuccessful treatment in comparison to those with homes (HR: 3.54, 95% CI 3.18–3.94, *p* = <0.001), even after controlling for other

**Table 1. Sociodemographic and clinical characteristics of tuberculosis patients in Brazil (n = 284,874; Year: 2015–2020).**

| | Sheltered population | | Homeless population | | P -value (α = 0.05) |
|---|---|---|---|---|---|
| | n = 277,125 | % = 97.28 | n = 7,749 | % = 2.72 | |
| **Sex** | | | | | <0.001* |
| Female | 84,548 | 30.51 | 1,301 | 16.79 | |
| Male | 192,561 | 69.48 | 6,448 | 83.21 | |
| Missing | 16 | 0.01 | - | - | |
| **Race** | | | | | <0.001 |
| White | 79,981 | 28.86 | 1,805 | 23.29 | |
| Black or mixed ethnicity | 155,793 | 56.22 | 4,583 | 59.14 | |
| Asian | 1,625 | 0.59 | 47 | 0.61 | |
| Indigenous | 2,433 | 0.87 | 38 | 0.49 | |
| Missing | 37,293 | 13.46 | 1,276 | 16.47 | |
| **Age group (in years)** | | | | | <0.001 |
| 18–29 | 86,825 | 31.33 | 1,379 | 17.79 | |
| 30–39 | 60,271 | 21.75 | 2,441 | 31.50 | |
| 40–49 | 46,390 | 16.74 | 2,090 | 26.97 | |
| 50–59 | 40,138 | 14.48 | 1,293 | 16.69 | |
| 60–69 | 25,784 | 9.31 | 432 | 5.58 | |
| 70–90 | 17,717 | 6.39 | 114 | 1.47 | |
| **Levels of education** | | | | | <0.001 |
| Bachelor or higher levels | 7,944 | 2.87 | 79 | 1.02 | |
| Tertiary education | 23,568 | 8.50 | 487 | 6.29 | |
| Secondary education | 25,590 | 9.23 | 886 | 11.43 | |
| Illiterate | 1,410 | 0.51 | 53 | 0.68 | |
| Missing | 218,613 | 78.89 | 6,244 | 80.58 | |
| **Beneficiary of government cash transfer program** | | | | | <0.001 |
| No | 153,965 | 55.56 | 3,637 | 46.94 | |
| Yes | 14,456 | 5.22 | 383 | 4.94 | |
| Missing | 108,704 | 39.22 | 3,729 | 48.12 | |
| **Clinical features of tuberculosis** | | | | | <0.001 |
| Extra-pulmonary tuberculosis | 34,441 | 12.43 | 370 | 4.77 | |
| Pulmonary tuberculosis | 242,684 | 87.57 | 7.379 | 95.23 | |
| **Region of Brazil** | | | | | <0.001 |
| Southeast | 132,794 | 47.92 | 4,254 | 54.89 | |
| North | 32,825 | 11.84 | 465 | 6.00 | |
| Northeast | 65,644 | 23.69 | 1,287 | 16.61 | |
| Central west | 13,235 | 4.78 | 408 | 5.27 | |
| South | 32,627 | 11.77 | 1,335 | 17.23 | |
| **HIV** | | | | | <0.001 |
| No coinfection | 251,804 | 90.87 | 6,136 | 79.18 | |
| Coinfection | 24,370 | 8.79 | 1,579 | 20.38 | |
| Missing | 951 | 0.34 | 34 | 0.44 | |
| **Alcohol use** | | | | | <0.001 |
| No | 220,302 | 79.49 | 3,165 | 40.84 | |
| Yes | 45,743 | 16.51 | 4,205 | 54.27 | |
| Missing | 11,080 | 4.00 | 379 | 4.89 | |
| **Diabetes** | | | | | <0.001 |

(*Continued*)

**Table 1.** (Continued)

| | Sheltered population | | Homeless population | | P -value (α = 0.05) |
|---|---|---|---|---|---|
| No | 243,797 | 87.97 | 6,897 | 89.00 | |
| Yes | 22,351 | 8.07 | 305 | 3.94 | |
| Missing | 10,977 | 3.96 | 547 | 7.06 | |
| **Mental disorder** | | | | | <0.001 |
| No | 259,806 | 93.75 | 6,698 | 86.44 | |
| Yes | 5,895 | 2.13 | 472 | 6.09 | |
| Missing | 11,424 | 4.12 | 579 | 7.47 | |
| **Tobacco use** | | | | | <0.001 |
| No | 202,094 | 72.93 | 3,587 | 46.29 | |
| Yes | 63,546 | 22.93 | 3,661 | 47.24 | |
| Missing | 11,485 | 4.14 | 501 | 6.47 | |
| **Illicit drug use** | | | | | <0.001 |
| No | 229,485 | 82.81 | 3,163 | 40.82 | |
| Yes | 34,909 | 13.00 | 4,109 | 53.03 | |
| Missing | 12,731 | 4.59 | 477 | 6.15 | |
| **Directly observed treatment-DOT** | | | | | 0.001 |
| No | 105,802 | 38.18 | 2,898 | 37.39 | |
| Yes | 106,481 | 38.42 | 2,887 | 37.26 | |
| Missing | 64,842 | 23.40 | 1,964 | 25.35 | |
| **Treatment outcomes** | | | | | <0.001* |
| Treatment success | 209,146 | 75.47 | 3,086 | 39.84 | |
| Loss to follow-up | 29,950 | 10.81 | 2,791 | 36.02 | |
| Death | 22,841 | 8.24 | 1,114 | 14.37 | |
| Treatment failure | 166 | 0.06 | 5 | 0.06 | |
| Not evaluated | 15,022 | 5.42 | 753 | 9.71 | |

*Fisher exact test.

covariates in the analysis (HR 2.04, 95% CI 1.82–2.28, $p$ = <0.001). Table 3 shows the association between socio-demographic and clinical covariates and the risk of unsuccessful treatments. Among the sheltered population, the risk of unsuccessful TB treatment is higher among men (HR 1.13, 95% CI 1.05–1.21, $p$ = <0.001), black or mixed ethnicity (HR 1.17, 95% CI 1.10–1.24, $p$ = <0.001), and indigenous population (HR 1.90, 95% CI 1.17–3.07, $p$ = 0.001).

**Table 2. Association of Homelessness with Unsuccessful Treatment Outcomes Populations in Brazil (2015–2020).**

| | Treatment Outcomes n(%) | | Hazard Ratio | | | |
|---|---|---|---|---|---|---|
| | Successful | Un-successful | Crude HR (95%CI) | p -value | Adjusted HR (95%CI) | p -value |
| Shelthered populations (n = 31830) | 27451 (86.2) | 4379 (13.8) | Reference | <0.001 | Reference | <0.001 |
| **Homeless populations (n = 8830)** | 464 (55.8) | 368(44.2) | 3.54 (3.18–3.94) | | 2.04 (1.82–2.28) | |
| | 27915 | 4747 | | | | |

HR: Hazard Ratio; CI: Confidence Interval; n: number of outcomes.

For age, the risk increases proportionally with age, and those above age 70 had the highest risk (HR 2.29, 95% CI 1.99–2.65, $p$ = <0.001). In contrast, the risk reduces with level of education, with the highest risk observed among the illiterates (HR: 1.85, 95% CI 1.51–2.26, $p$ = <0.001). Also, the practice of not performing a DOT is related to unsuccessful treatment, with those that are not monitored been 16.18 times less likely to complete a successful course of treatment (95% CI 14.67–17.84, $p$ = <0.001), when compared to those who get accompanied. However, none of the sociodemographic covariates were associated with the risk of unsuccessful treatments among the HP (Table 3).

Furthermore, the risk of not completing a successful course of TB treatment was higher among sheltered population living with HIV (HR 1.98; 95% CI 1.82–2.15, $p$ = <0.001), users of illicit drugs (HR 1.75; 95% CI 1.62–1.88, $p$ = <0.001), people with diabetes (HR 1.04; 95% CI 0.98–1.11, p = 0.01) and those who consumes alcohol (HR 1.30; 95% CI 1.21–1.40, $p$ = <0.001). Similarly, these clinical covariates were significantly associated with risk of having an unsuccessful TB treatment among the HP. Patients who use illicit drugs (HR 1.38, 95% CI 1.09–1.74, $p$ = 0.001), had mental disorders (HR 2.12, 95% CI 1.08–4.15, $p$ = 0.01) or have not received DOT (HR 18.37, 95% CI 12.23–27.58, $p$ = <0.001) have higher risk of an unsuccessful TB treatment. The model, however, shows that those who consumes alcohol are less likely to have unsuccessful TB treatment (HR 0.77, 95% CI 0.61–0.98, $p$ = 0.01) (Table 3).

## Factors associated with undesirable treatment outcomes among homeless population

Table 4 shows the association between the three covariates that increases the risk of unsuccessful treatment among the HP, and their contribution to undesirable treatment outcomes such as loss to follow-up and death. All the three covariates were significantly associated with loss to follow-up events, with the absence of DOT having the highest effect on loss to follow-up (HR 17.97, 95% CI 11.71–27.59, $p$ = <0.001), followed by mental disorder (HR 2.24, 95% CI 1.11–4.55, $p$ = 0.01) and use of illicit drug use (HR 1.53, 95% CI 1.21–1.93, $p$ = <0.001). However, only the absence of DOT influenced mortality (HR 15.66, 95% CI 4.79–51.15, $p$ = <0.001).

## Discussion

This study found some critical issues impeding successful completion of TB treatment among HP, which ultimately increases their risk and burden of the disease experienced. Our analysis show that TB affects more than half of the patients who identified as black or of mixed ethnicity (59.14% of homeless patients and 56.22% of sheltered patients) and reiterates earlier notions that skin color is one of the major determinants of TB risk [23]. The association between skin color and unsuccessful treatment outcome in TB is likely influenced by social factors rather than biological ones. Various studies have highlighted the challenges faced by individuals of black or mixed ethnicity in relation to TB. These challenges include inadequate access to appropriate treatment, higher levels of unemployment, poor housing and transportation conditions, and cultural barriers such as health literacy, beliefs, and stigma related to TB within specific populations [24–30]. Black populations from Brazil, have also been reported to have a lower level of education [31,32,33], which has a direct impact on their socio-economic status and increases their vulnerability to tuberculosis (TB) and other infectious diseases associated with poverty. It is therefore crucial to recognize the factors that contribute to the social context, leading to disparities in tuberculosis (TB) treatment outcomes among individuals from specific ethnic backgrounds. The understanding of these social determinants can help inform interventions and policies aimed at addressing the root causes of such disparities and promoting equitable TB care for all populations. Optimizing healthcare interventions that

**Table 3. Association between sociodemographic and clinical covariates and the risk of unsuccessful TB treatments among sheltered and homeless populations in Brazil (2015–2020).**

| | Sheltered Population (n = 31,830) | | | | Homeless population (n = 832) | | | |
|---|---|---|---|---|---|---|---|---|
| | n | Crude HR (95%CI) | Adjusted HR (95%CI) | P -value | n | Crude HR (95%CI) | Adjusted HR (95%CI) | P -value |
| **Sex** | | | | | | | | |
| Female | 8090 | Reference | Reference | | 128 | Reference | Reference | |
| Male | 23740 | 1.23 (1.15–1.32) | 1.13 (1.05–1.21) | <**0.001** | 704 | 0.72 (0.55–0.93) | 1.25 (0.94–1.67) | 0.11 |
| **Race** | | | | | | | | |
| White | 14294 | Reference | Reference | | 273 | Reference | Reference | |
| Black or mixed ethnicity | 17402 | 1.17 (1.10–1.24) | 1.17 (1.10–1.24) | <**0.001** | 557 | 1.02 (0.82–1.27) | 1.18 (0.95–1.48) | |
| Asian | 49 | 1.68 (0.90–3.13) | 1.47 (0.78–2.74) | 0.22 | 1 | 31.92 (4.30–236.56) | 27.76 (3.52–218.84) | <**0.001** |
| Indigenous | 85 | 1.51 (0.94–2.44) | 1.90 (1.17–3.07) | **0.001** | 1 | 0.00 (0.00 –Inf) | 0.00 (0.00 –Inf) | 0.99 |
| **Age group** | | | | | | | | |
| 18–29 | 12282 | Reference | Reference | | 127 | Reference | Reference | |
| 30–39 | 7561 | 1.17 (1.09–1.26) | 1.05 (0.98–1.14) | 0.14 | 281 | 0.92 (0.69–1.22) | 1.06 (0.79–1.42) | 0.67 |
| 40–49 | 5008 | 1.23 (1.13–1.34) | 1.06 (0.97–1.15) | 0.19 | 231 | 0.69 (0.51–0.94) | 0.87 (0.63–1.18) | 0.38 |
| 50–59 | 3727 | 1.10 (1.00–1.22) | 1.13 (1.02–1.25) | **0.01** | 137 | 0.35 (0.23–0.54) | 0.57 (0.37–0.87) | **0.01** |
| 60–69 | 2147 | 1.30 (1.15–1.46) | 1.46 (1.29–1.65) | <**0.001** | 51 | 0.48 (0.28–0.83) | 0.92 (0.52–1.63) | 0.79 |
| 70–90 | 1105 | 1.94 (1.70–2.22) | 2.29 (1.99–2.65) | <**0.001** | 5 | 1.42 (0.44–4.53) | 1.73 (0.52–5.69) | 0.36 |
| **Education level** | | | | | | | | |
| Bachelor or higher levels | 3383 | Reference | Reference | | 33 | Reference | Reference | |
| Tertiary education | 12880 | 1.59 (1.41–1.81) | 1.44 (1.27–1.63) | <**0.001** | 260 | 0.73 (0.44–1.22) | 0.65 (0.38–1.10) | 0.11 |
| Secondary education | 14784 | 1.99 (1.77–2.25) | 1.61 (1.42–1.82) | <**0.001** | 507 | 0.83 (0.50–1.36) | 0.71 (0.42–1.18) | 0.19 |
| Illiterate | 783 | 2.45 (2.02–2.97) | 1.85 (1.51–2.26) | <**0.001** | 32 | 0.87 (0.43–1.74) | 1.13 (0.55–2.30) | 0.73 |
| **Clinical disease status** | | | | | | | | |
| Extra-pulmonary tuberculosis | 3539 | Reference | Reference | | 23 | Reference | Reference | |
| Pulmonary tuberculosis | 28291 | 1.13 (1.02–1.24) | 1.04 (0.95–1.15) | 0.35 | 809 | 1.17 (0.57–2.16) | 1.08 (0.54–2.13) | 0.81 |
| **HIV** | | | | | | | | |
| No coinfection | 29956 | Reference | Reference | | 706 | Reference | Reference | |
| Coinfection | 1874 | 2.74 (2.53–2.98) | 1.98 (1.82–2.15) | <**0.001** | 126 | 1.63 (1.25–2.09) | 1.16 (0.88–1.53) | 0.26 |
| **Alcohol use** | | | | | | | | |
| No | 26168 | Reference | Reference | | 352 | Reference | Reference | |
| Yes | 5662 | 1.91 (1.80–2.04) | 1.30 (1.21–1.40) | <**0.001** | 480 | 0.88 (0.71–1.08) | 0.77 (0.61–0.98) | **0.01** |
| **Diabetes mellitus** | | | | | | | | |
| No | 29926 | Reference | Reference | | 804 | Reference | Reference | |
| Yes | 1904 | 1.12 (1.00–1.26) | 1.04 (0.98–1.11) | **0.01** | 28 | 0.88 (0.49–1.57) | 0.78 (0.42–1.42) | 0.42 |
| **Tobacco use** | | | | | | | | |
| No | 23166 | Reference | Reference | | 490 | Reference | Reference | |
| Yes | 8664 | 1.36 (1.28–1.44) | 1.04 (0.98–1.11) | 0.17 | 342 | 1.03 (0.84–1.27) | 0.95 (0.76–1.18) | 0.67 |
| **Illicit drugs use** | | | | | | | | |
| No | 26003 | Reference | Reference | | 385 | Reference | Reference | |
| Yes | 5827 | 2.10 (1.98–2.24) | 1.75 (1.62–1.88) | <**0.001** | 447 | 1.52 (1.23–1.87) | 1.38 (1.09–1.74) | **0.001** |
| **Mental disorder** | | | | | | | | |
| No | 31401 | Reference | Reference | | 812 | Reference | Reference | |
| Yes | 429 | 1.17 (0.93–1.47) | 1.02 (0.81–1.28) | 0.85 | 20 | 1.11 (0.57–2.15) | 2.12 (1.08–4.15) | **0.01** |
| **Directly observed treatment (DOT)** | | | | | | | | |
| Yes | 18938 | Reference | Reference | | 356 | Reference | Reference | |
| No | 12892 | 16.72 (15.17–18.44) | 16.18 (14.67–17.84) | <**0.001** | 476 | 18.05 (12.08–26.95) | 18.37 (12.23–27.58) | <**0.001** |

Model adjusted for all covariates; HR: Hazard Ratio; CI: Confidence Interval; n: number of outcomes.

**Table 4. Factors associated with undesirable treatment outcomes among homeless populations in Brazil (n = 830*; Year: 2015–2020).**

| | Loss to Follow-Up | | | Death | | |
|---|---|---|---|---|---|---|
| | Crude HR (95%CI) | Adjusted HR (95%CI) | p-value | Crude HR (95%CI) | Adjusted HR (95%CI) | p-value |
| **Illicit drugs uses** | | | | | | |
| No | Reference | Reference | | Reference | Reference | |
| Yes | 1.75 (1.39–2.21) | 1.53 (1.21–1.93) | **<0.001** | 0.69 (0.38–1.25) | 0.60 (0.32–1.09) | 0.09 |
| **Mental disorder** | | | | | | |
| No | Reference | Reference | | Reference | Reference | |
| Yes | 1.15 (0.57–2.33) | 2.24 (1.11–4.55) | **0.01** | 1.02 (0.14–7.45) | 1.59 (0.21–11.69) | 0.64 |
| **Directly observed treatment (DOT)** | | | | | | |
| Yes | Reference | Reference | | Reference | Reference | |
| No | 18.02 (11.76–27.61) | 17.97 (11.71–27.59) | **<0.001** | 14.9 (4.57–48.52) | 15.67 (4.79–51.15) | **<0.001** |

HR: Hazard Ratio; CI: Confidence Interval

*2 records from the Asian and indigenous categories under the variable "Race" were further excluded from the homeless population dataset due to very few observations making the total records 830.

protects neglected black populations from infectious diseases like TB, may also have indirect economic benefits [20].

Furthermore, our findings shows that HP had higher loss to follow-up rates and death in comparison to sheltered populations. The lost to follow-up rates and deaths observed in our study are higher than those reported recently in another cross-sectional survey in Brazil, where about 28.8% homeless persons were loss to follow-up and 8.1% of them died [5]. The homeless populations were also less likely to complete successful course of TB treatments when compared with sheltered ones, and the risk were increased when they are of the black or mixed ethnicity, had HIV co-infection, and uses illicit drugs. This finding corroborates other reports in Brazil, where homeless persons were generally 5 times less likely to complete treatment, and those who uses alcohol and illicit drugs were twice less likely to complete treatment [6]. Another study in São Paulo, also found a similar result, where HP were twice less likely to complete treatment, and those who were black, co-infected with HIV, uses alcohol or illicit drugs shared lower risk of completing TB treatment [34]. Other study from the southern region of the country also shows that over 23% of people living on the street use illicit drugs and abandon TB treatments [35]. However, our findings show that HP who uses alcohol were less likely to have unsuccessful TB treatments, which contrast with other studies that have highlighted alcohol as a determinant of worst TB outcome [6,22,36]. Perhaps, these population access different health-care service due to alcohol issue, which may be helpful to treat TB. However, there were no data to substantiate this. Nevertheless, the findings from this study emphasize that homelessness, skin color, HIV comorbidity and use of illicit drugs are critical factors that needs to be factored into current prevention and control strategies for TB.

Furthermore, we observed that the lack of DOT has a very strong effect on the likelihood of unsuccessful treatments, and in fact undesirable outcomes such as loss to follow-ups and deaths. It is therefore important that efforts targeted at improving treatment success must incorporate innovative strategies of performing DOT among the homeless population. This would require investments in resources at institutional and social levels. As previously described by Maciel and Reis-Santos [22], the TB- risk determinant framework is complex, and efforts targeted at controlling the diseases especially among HP would require a multisectoral and dynamic approach. Foremost, it would be imperative to consider how social determinants and biosocial characteristics of individuals influences TB treatments and outcomes [37].

This should then be accompanied by inter-sectoral actions that addresses and provides sustainable alternatives to the psychological, financial, food insecurity and health problems of the homeless populations [37–39]. As such, integration among public institutions that take care of vulnerable people become essential. Previous studies of Alecrim et al. [37] among health-care workers and HP in Sao Paulo, found that provision of transport vouchers and restrooms at the health-care units encouraged adherence of HP to TB treatment, which allowed better practice of DOT. Also, provision of foods was also found to be interesting, but with concerns that participants may purposively miss TB treatment appointment due to the fear of suspending the food support after completion of treatment [37]. It is therefore important to carefully identify potential solutions that encourages participation and retention of HP in TB treatments.

One potential strategy to support retention of homeless population in TB treatment programs is to reinforce the concept of the 'Street Clinic', known as *Consultório na rua–CnaR*. The *CnaR* is a government policy for PHC, which was modelled after the work against drugs abuse and mental illness in Salvador-BA since 1997 and has become an expansion of the SUS in Brazil that assists HP[38]. The *CnaR* workers may aid the process of identifying TB patients and delivering care to these homeless people [39]. However, the PHC in Brazil has faced significant financing constraints regarding the implementation of this policy. The current allocation of budget is contingent on performance evaluation, which directly impacts the increase in job demands placed on employees to meet predetermined targets. However, this policy has resulted in a decrease in the quality of work among healthcare practitioners, potentially leading to unimproved health outcomes. [40]. The health workers employed within the *CnaR* may therefore be responsible for achieving institutional goals, but may encounter challenges in performing their duties effectively, leading to a disparity between the expected and actual performance levels. Moreover, other factors such as reduction of investments in SUS, the constitutional amendment, austerity measures promoted by (neo)liberal governments since 2016 and privatization of health through social organizations [41,42] have impeded the proper functioning of public health care system in Brazil and, consequently, people's health condition.

In summary, our study has significant public health implications for TB control in Brazil, by highlighting social factors impacting TB treatment outcomes among neglected populations. Specifically, we found that homelessness, being of black or mixed ethnicity, lack of Directly Observed Therapy (DOT), HIV co-infection, and drug use were associated with higher rates of loss to follow-up and death. These findings underscore the need to address the specific challenges limiting the adherence of homeless populations to TB and HIV diagnosis and care interventions. It is also crucial to implement targeted interventions aimed at mitigating the socio-economic vulnerabilities experienced by individuals of black or mixed ethnicity, including improving access to education, employment opportunities, and housing conditions. Investments that strengthen integration among public institutions that provide support for vulnerable populations, such as the Consultório na rua (CnaR) are also very essential. Employing a multi-sectoral approach, that considers these social determinants of health, and other psychological, financial, and food insecurity issues faced by these populations can improve the retention and adherence of vulnerable populations to TB treatment, ultimately leading to better TB control outcomes, and presumably those of other infectious diseases in Brazil.

This study has some limitations. Foremost is the large number of missing values which led to the omission of some records in our final models. We believe the characteristics of the records omitted might further enrich our analysis. It is therefore important to invest efforts in ensuring synergies and closing gaps that exist in data reportage among all the varying sources that feeds the SINAN. Secondly, access to safe and nutritious food is an important factor that influences clinical conditions of TB patients, and HP are mostly in need of sufficient and nourishing meals [37]. These population are most likely to prioritize acquisition of food and shelter

over other health-seeking behavior such as TB treatment [43]. As such, if HP have these primary necessities met, it could better facilitate their willingness to complete TB treatment [43]. However, it's important to note that our models considered the covariates available in the SINAN database. These covariates were mostly dichotomized, which could have limited the variability in the data. Furthermore, the covariate on nutritional status or access to food supply was not captured on the database. Thirdly, the notification end date does not necessarily coincide with the treatment end date, but the date the case was closed on SINAN. Thus, the use of the closing date can lead to interpretations that do not necessarily reflect the duration of treatment. To reduce doubts, we performed the sensitivity analyses to confirm if the results would be the similar over different periods. Fourthly, some TB deaths may be registered only at the Mortality Information System (*Sistema de Informação sobre Mortalidade*, SIM), and since we did not link both systems (i.e., SINAN and SIM), some of the mortality data might have been underestimated. Finally, since TB is an infectious disease related to poverty [20], and most HP receive lower or no income [6], investigating the impact of cash transfer programs could better highlight the influence of social support policies on TB treatments and outcomes. Unfortunately, we couldn't explore this association because of the large amounts of missing data in our dataset. However, previous studies by Andrade et al. [33], found a higher proportion of TB treatment were observed among people that received government aids. Since more than 45% of the HP in our study are non-beneficiaries of cash-transfer programs, it is therefore important for government to invest in more innovative and inclusive strategies that increases the coverage of financial aids to these vulnerable population.

## Conclusion

This study revealed that the HP in Brazil are at risk of not completing the standard course of treatment for TB, and more particularly among those who use illicit drugs, have mental illnesses or not been monitored via DOT for TB. Furthermore, the lack of DOT has a very strong effect on the likelihood of undesirable outcomes such as loss to follow-ups and deaths. It is therefore important that efforts targeted at improving treatment success must incorporate innovative strategies of performing DOT and encouraging participation and retention of HP in TB treatment programs.

## Supporting information

**S1 Fig. Conceptual framework for tuberculosis showing the sociodemographic and clinical covariates included in the predictive model.**
(TIF)

**S2 Fig. Flowchart of the study.** (Data source: SINAN—tuberculosis updated in August 2021).
(TIF)

**S3 Fig. Cumulative incidence curve of unsuccessful TB treatment among homeless and sheltered population in Brazil (2015–2020).**
(TIF)

**S1 File Analysis Code. R Script.**
(R)

**S1 File Data frame. CSV data frame.**
(CSV)

**S1 Table. Descriptive sensitivity analyses by comparing treatment periods of 180 days.**
(DOCX)

**S2 Table. Descriptive sensitivity analyses by comparing treatment periods of 360 days.** (DOCX)

## Acknowledgments

We kindly say thank you to Andrêa Sheila Ferreira, who supported the corresponding author with the conceptual predictive model (S1 Fig) and suggested social covariates for the analysis.

## Author Contributions

**Conceptualization:** Osiyallê Akanni Silva Rodrigues, Layana Costa Alves, Cleber Cremonese, Joilda Silva Nery.

**Data curation:** Osiyallê Akanni Silva Rodrigues, Renzo Flores-Ortiz, Cleber Cremonese.

**Formal analysis:** Osiyallê Akanni Silva Rodrigues, Hammed Oladeji Mogaji, Renzo Flores-Ortiz.

**Investigation:** Osiyallê Akanni Silva Rodrigues, Hammed Oladeji Mogaji, Layana Costa Alves, Renzo Flores-Ortiz, Cleber Cremonese, Joilda Silva Nery.

**Methodology:** Osiyallê Akanni Silva Rodrigues, Layana Costa Alves, Renzo Flores-Ortiz, Cleber Cremonese, Joilda Silva Nery.

**Project administration:** Cleber Cremonese, Joilda Silva Nery.

**Resources:** Osiyallê Akanni Silva Rodrigues, Layana Costa Alves, Cleber Cremonese, Joilda Silva Nery.

**Software:** Osiyallê Akanni Silva Rodrigues, Renzo Flores-Ortiz.

**Supervision:** Renzo Flores-Ortiz, Cleber Cremonese, Joilda Silva Nery.

**Validation:** Cleber Cremonese, Joilda Silva Nery.

**Visualization:** Osiyallê Akanni Silva Rodrigues, Hammed Oladeji Mogaji, Layana Costa Alves, Renzo Flores-Ortiz.

**Writing – original draft:** Osiyallê Akanni Silva Rodrigues, Layana Costa Alves, Renzo Flores-Ortiz, Cleber Cremonese, Joilda Silva Nery.

**Writing – review & editing:** Osiyallê Akanni Silva Rodrigues, Hammed Oladeji Mogaji, Layana Costa Alves, Renzo Flores-Ortiz, Cleber Cremonese, Joilda Silva Nery.

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
