## [Decision Letter · Decision Letter 0]

30 Jan 2023

Dear Dr. Rodrigues,

Thank you very much for submitting your manuscript "Factors associated with unsuccessful tuberculosis treatment among homeless persons in Brazil: a retrospective cohort study from 2015 to 2020" for consideration at PLOS Neglected Tropical Diseases. As with all papers reviewed by the journal, your manuscript was reviewed by members of the editorial board and by several independent reviewers. In light of the reviews (below this email), we would like to invite the resubmission of a significantly-revised version that takes into account the reviewers' comments. 

We cannot make any decision about publication until we have seen the revised manuscript and your response to the reviewers' comments. Your revised manuscript is also likely to be sent to reviewers for further evaluation.

Sincerely,

Subash Babu

Academic Editor

Ana LTO Nascimento

Section Editor

Reviewer's Responses to Questions

**Key Review Criteria Required for Acceptance?**

**Methods**

-Are the objectives of the study clearly articulated with a clear testable hypothesis stated?

-Is the study design appropriate to address the stated objectives?

-Is the population clearly described and appropriate for the hypothesis being tested?

-Is the sample size sufficient to ensure adequate power to address the hypothesis being tested?

-Were correct statistical analysis used to support conclusions?

-Are there concerns about ethical or regulatory requirements being met?

Reviewer #1: This study tries to compare the TB treatment outcomes between patients who are homeless with those not homeless. The design and the groups included in the study are appropriate. However, the objectives though scientifically valid, need to be stated using language that is clear. Sample size has not been calculated for this study. It will be good to calculate and see whether the numbers finally available for complete analysis are enough to draw valid conclusions. It is understood that the database represents all TB patients in the country. But, there are large numbers with missing values. Therefore it will be good to check the characteristics of those omitted from the analysis with those finally included. It is possible that the patients who got included were not representative of all TB patients.

Statistical analysis is appropriate for the stated objectives.

Reviewer #2: Line 133: Were transferred to other units, unknown cases and missing values included under unsuccessful treatment?

Line 169: Selecting independent variables which were significant for composite unsuccessful outcome and applying to individual categories of outcome will not be accurate. All relevant independent variables must be included in the adjusted analysis for unsuccessful outcomes.

**Results**

-Does the analysis presented match the analysis plan?

-Are the results clearly and completely presented?

-Are the figures (Tables, Images) of sufficient quality for clarity?

Reviewer #1: The analysis is appropriately done and results represented in tables. However, the table titles need to be modified in more intelligible English.

Figures need to competed and self-explanatory with titles and foot notes.

Reviewer #2: 1. Table 1: Alcohol misuse- Does this population include those who consume alcohol or those with alcohol abuse/misuse. How was it measured during original data collection. Variable could be renamed as alcohol use rather than misuse.

2. Table 1: Numbers with no alcohol use is wrong.

3. Line 207: The term cured is used interchangeably with treatment success which is not appropriate.

4. Line 218: Unsuccessful treatment is mentioned as unconcluded treatment. Unconcluded treatment usually means incomplete treatment. It is not appropriate to use this term in this context.

5. Line 239: The Hazard ratio mentioned is wrongly interpreted

6. Table 3: The n mentioned here is 830; in flowchart it is mentioned as 832.

**Conclusions**

-Are the conclusions supported by the data presented?

-Are the limitations of analysis clearly described?

-Do the authors discuss how these data can be helpful to advance our understanding of the topic under study?

-Is public health relevance addressed?

Reviewer #1: The analysis is appropriate, but does not include many variables that can influence the TB treatment outcomes, such as nutritional status and severity of disease. Information on these variables is difficult to obtain from country wide databases. But this can be one reason why alcohol use was found to be protective against adverse outcomes. Missing data is a major issue which has been discussed, but no attempt made to identify or quantify its effect on the effect measures.

Reviewer #2: The discussion and inferences from the study should be better articulated.

**Editorial and Data Presentation Modifications?**

Reviewer #1: Overall, the language needs modification, to be presented in more intelligible fashion.

Reviewer #2: Typographical and grammatical errors are seen throughout the manuscript. The language is unclear, making it difficult to follow. The manuscript should be edited to improve the flow and readability of the text.

**Summary and General Comments**

Reviewer #1: This is an important study highlighting the failure of TB treatment for the most vulnerable section of the society, the homeless. This section needs to be focussed for TB programme to succeed. The study utilises country wide database, which has both advantages and disadvantages. The representativeness is good, but at the same time, large number of missing values reduces validity of the results.

Reviewer #2: The authors report the risk factors associated with unfavorable treatment outcomes among homeless population which is one of the neglected study group. It is a good attempt by the author to include data over a period of five years. The appropriateness of statistical analysis performed (Cox regression analysis) and their interpretations require more scrutiny by a biostatistician. The discussion and inferences from the study should be articulated in a better manner. The discussion can be made more succinct. Typographical and grammatical errors are seen throughout the manuscript; grammatical errors in few areas seem to change the interpretation of the text.

PLOS authors have the option to publish the peer review history of their article (what does this mean?). If published, this will include your full peer review and any attached files.

Reviewer #1: No

Reviewer #2: No
---

## [Decision Letter · Decision Letter 1]

29 May 2023

Dear Dr. Rodrigues,

Thank you very much for submitting your manuscript "Factors associated with unsuccessful tuberculosis treatment among homeless persons in Brazil: a retrospective cohort study from 2015 to 2020" for consideration at PLOS Neglected Tropical Diseases. As with all papers reviewed by the journal, your manuscript was reviewed by members of the editorial board and by several independent reviewers. The reviewers appreciated the attention to an important topic. Based on the reviews, we are likely to accept this manuscript for publication, providing that you modify the manuscript according to the review recommendations. 

Sincerely,

Subash Babu

Academic Editor

Ana LTO Nascimento

Section Editor

Reviewer's Responses to Questions

**Key Review Criteria Required for Acceptance?**

**Methods**

-Are the objectives of the study clearly articulated with a clear testable hypothesis stated?

-Is the study design appropriate to address the stated objectives?

-Is the population clearly described and appropriate for the hypothesis being tested?

-Is the sample size sufficient to ensure adequate power to address the hypothesis being tested?

-Were correct statistical analysis used to support conclusions?

-Are there concerns about ethical or regulatory requirements being met?

Reviewer #1: Introduction: There are multiple studies on TB among homeless people, also from Brazil. How does this add to the available knowledge should be added in the justification for the study. There is a description of the homeless people but, not on the burden of TB among the homeless people, critical appraisal of the evidences available and the existing gaps in knowledge can be added in the introduction.

Study setting: Some details regarding the TB treatment in Brazil, such as 1) the treatment following alternate day or daily dose regimen and 2) whether DOTS is offered to all TB patients in the country, will be helpful for the readers to understand the context.

Exclusion criteria: The statement is not clear as to what are the stages being referred to.

Outcome variable: 'Loss to follow-up' (LTFU) and 'not evaluated' seem to be similar for research purpose as the treatment outcome could not have been assessed for either categories. Were these two groups combined in the analysis?

Covariates: Severity of alcohol and tobacco use and control status of diabetes would have an effect on the treatment outcomes. Dichotomisation of these variables reduces their influence on the outcomes.

Statistical analysis:

1) The covariates included in the model for assessing the association of specific risk factors with LTFU and death, displayed in table 3, are not mentioned.

2) Lines 173, 174 and 175: Points two and three appear to be same.

3) It is mentioned that sensitivity analysis was performed to see the effect of varying durations of treatment periods. However, the findings of the sensitivity analysis is not described in the result section.

Reviewer #3: The purpose of the study was to evaluate factors associated with unsuccessful TB treatment and identify the risk factors for undesirable TB treatment outcomes among homeless population in Brazil in comparison to sheltered population. 

The study background and introduction are clear with appropriate references from literature.

The study is appropriately designed and sample size determination is clearly explained for the study groups. Study population, data source and data collection methodology, outcome variables, covariates are mentioned in detail. The study analysis is appropriate for the study aims.

No ethical or regulatory concerns were noted.

**Results**

-Does the analysis presented match the analysis plan?

-Are the results clearly and completely presented?

-Are the figures (Tables, Images) of sufficient quality for clarity?

Reviewer #1: Table no 2: How can homelessness be a risk factor for the sheltered group of patients?

For tables 2 and 3, only the HR are given for the covariates. The number of outcomes in the various categories of the covariates will add more information.

Reviewer #3: The study analysis is appropriate. However, few discrepancies were noted in the stated numbers.

In Table 1, n=277, 125 for sheltered population but the total counts to 282,125 for the socio-demographic covariate ‘Region of Brazil’’.

In Table 2, p-value is missing for Homeless population for the variable ‘Black or mixed ethnicity’.

At line 240, p-value for indigenous population is mentioned as <0.001 but is mentioned as 0.001 in Table 2. 

At line 260, p-value mentioned for alcohol use among homeless population is mentioned as 0.03 whereas in Table 2, p-value is mentioned as 0.01.

In the S1 Supporting Information, n=89,987 for sheltered population. However, the total counts to 90,057 for the covariate ‘Skin colour’.

**Conclusions**

-Are the conclusions supported by the data presented?

-Are the limitations of analysis clearly described?

-Do the authors discuss how these data can be helpful to advance our understanding of the topic under study?

-Is public health relevance addressed?

Reviewer #1: Skin colour as a risk factor for unsuccessful treatment outcome is probably associated with social rather than biological factors. This point could be discussed more. 

Public health implications of the findings of this study vis-a-vis TB control in Brazil is not effectively discussed.

Reviewer #3: No comments for the details provided under study limitations and conclusions.

**Editorial and Data Presentation Modifications?**

Reviewer #1: The data presentation has been modified. Still, clarity is lacking in the way the tables are presented.

Reviewer #3: Reference to S3 appendix and S4 appendix needs to be corrected at line 152.

There are minor typographical errors in the revised document in table 3 (drogues), S1 and S2 supporting information (p-valor) etc.

**Summary and General Comments**

Reviewer #1: Language corrections are needed in multiple places in the manuscript. Public health implications of homelessness on TB control needs to be strengthened.

Reviewer #3: None.

PLOS authors have the option to publish the peer review history of their article (what does this mean?). If published, this will include your full peer review and any attached files.

Reviewer #1: No

Reviewer #3: No

Figure Files:

Data Requirements:

Reproducibility:

References

---

## [Decision Letter · Decision Letter 2]

7 Aug 2023

Dear Dr. Rodrigues,

Thank you very much for submitting your manuscript "Factors associated with unsuccessful tuberculosis treatment among homeless persons in Brazil: a retrospective cohort study from 2015 to 2020" for consideration at PLOS Neglected Tropical Diseases. As with all papers reviewed by the journal, your manuscript was reviewed by members of the editorial board and by several independent reviewers. The reviewers appreciated the attention to an important topic. Based on the reviews, we are likely to accept this manuscript for publication, providing that you modify the manuscript according to the review recommendations. 

Sincerely,

Subash Babu

Academic Editor

Ana LTO Nascimento

Section Editor

Reviewer's Responses to Questions

**Key Review Criteria Required for Acceptance?**

**Methods**

-Are the objectives of the study clearly articulated with a clear testable hypothesis stated?

-Is the study design appropriate to address the stated objectives?

-Is the population clearly described and appropriate for the hypothesis being tested?

-Is the sample size sufficient to ensure adequate power to address the hypothesis being tested?

-Were correct statistical analysis used to support conclusions?

-Are there concerns about ethical or regulatory requirements being met?

Reviewer #1: The methods section is appropriate. Most of the queries and comments have been answered adequately. Some of the issues that could not be changed have been mentioned in the manuscript as limitations.

Reviewer #3: No comments.

**Results**

-Does the analysis presented match the analysis plan?

-Are the results clearly and completely presented?

-Are the figures (Tables, Images) of sufficient quality for clarity?

Reviewer #1: The results match the stated plan of analysis and are as per the objectives of the study. However, the language of presentation of the results need to be corrected for grammatical errors and also tense at various parts of the result section.

In response to one of the comment of the reviewers, the word 'cured' was replaced by 'healed', which is not appropriate as the tubercular lesions often do not heal completely with post-TB lung sequelae being an accepted entity. The term 'successful treatment' as stated in the operational definitions of the methods section can be used consistently.

Tables have many empty rows and columns of cells. Merging of cells can be done to make the tables neater. 

In table no.2, models have been separately done for homeless and sheltered population. How can there be homelessness in the reference category for the sheltered population and a reference category of sheltered population in the model for HP?

Reviewer #3: In the S1 Supporting information, n=89,987 for sheltered population. The total for the covariate 'skin colour' (89,977) still does not tally. Kindly confirm the correct values.

Also, kindly format Table 2 to fit the contents. The last 2 columns of the table are visible in the word document (Revised manuscript 16.07(track copy)) but has got truncated in the pdf.

**Conclusions**

-Are the conclusions supported by the data presented?

-Are the limitations of analysis clearly described?

-Do the authors discuss how these data can be helpful to advance our understanding of the topic under study?

-Is public health relevance addressed?

Reviewer #1: The findings of this study have public health relevance and the conclusions are supported by the data. Limitations have been added as per the suggestions of the reviewers.

Reviewer #3: No comments

**Editorial and Data Presentation Modifications?**

Reviewer #1: The language for this paper needs to be improved. There are grammatical errors through out the manuscript, more in the result section. Table no 2 needs correction as mentioned above for the results.

Reviewer #3: No comments

**Summary and General Comments**

Reviewer #1: This study has public health relevance even though there are some major limitations, which are due to the data source and not the design. Writing especially the grammar needs improvement. Tables need correction.

Reviewer #3: No major revisions. All observations have been addressed except comment 6.

PLOS authors have the option to publish the peer review history of their article (what does this mean?). If published, this will include your full peer review and any attached files.

Reviewer #1: No

Reviewer #3: No

Figure Files:

Data Requirements:

Reproducibility:

References

---

## [Decision Letter · Decision Letter 3]

13 Sep 2023

Dear Dr. Rodrigues,

Thank you very much for submitting your manuscript "Factors associated with unsuccessful tuberculosis treatment among homeless persons in Brazil: a retrospective cohort study from 2015 to 2020" for consideration at PLOS Neglected Tropical Diseases. As with all papers reviewed by the journal, your manuscript was reviewed by members of the editorial board and by several independent reviewers. The reviewers appreciated the attention to an important topic. Based on the reviews, we are likely to accept this manuscript for publication, providing that you modify the manuscript according to the review recommendations. 

Sincerely,

Subash Babu

Academic Editor

Ana LTO Nascimento

Section Editor

Reviewer's Responses to Questions

**Key Review Criteria Required for Acceptance?**

**Methods**

-Are the objectives of the study clearly articulated with a clear testable hypothesis stated?

-Is the study design appropriate to address the stated objectives?

-Is the population clearly described and appropriate for the hypothesis being tested?

-Is the sample size sufficient to ensure adequate power to address the hypothesis being tested?

-Were correct statistical analysis used to support conclusions?

-Are there concerns about ethical or regulatory requirements being met?

Reviewer #1: (No Response)

Reviewer #3: No comments. All observations have been addressed.

**Results**

-Does the analysis presented match the analysis plan?

-Are the results clearly and completely presented?

-Are the figures (Tables, Images) of sufficient quality for clarity?

Reviewer #1: All the comments of the reviewers have been addressed in the revised manuscript. There is only a minor point. 

In table 2, results from analysis of two multivariable models have been included. The crude and adjusted HR for homelessness is assumed to be from a model developed with the total study population, which should be 32,662 (31,830 sheltered + 832 homeless). However, the number mentioned is n=331,830.

The remaining table are results from two multivariable models done separately for the sheltered (n=31,830) and homeless populations (832). It is leading to some confusion. It will be better if first row of results, which addresses the first objective of exploring the association of homelessness with the unsuccessful treatment outcomes is presented in a separate table. Table 2 can retain the the title and the contents presenting the association of the sociodemographic and clinical covariates with unsuccessful treatment among the sheltered and homeless population separately, the second objective.

Reviewer #3: No comments. All observations have been addressed.

**Conclusions**

-Are the conclusions supported by the data presented?

-Are the limitations of analysis clearly described?

-Do the authors discuss how these data can be helpful to advance our understanding of the topic under study?

-Is public health relevance addressed?

Reviewer #1: (No Response)

Reviewer #3: No comments.

**Editorial and Data Presentation Modifications?**

Reviewer #1: All the comments of the reviewers have been addressed in the revised manuscript. There is only a minor point. 

In table 2, results from analysis of two multivariable models have been included. The crude and adjusted HR for homelessness is assumed to be from a model developed with the total study population, which should be 32,662 (31,830 sheltered + 832 homeless). However, the number mentioned is n=331,830.

The remaining table are results from two multivariable models done separately for the sheltered (n=31,830) and homeless populations (832). It is leading to some confusion. It will be better if first row of results, which addresses the first objective of exploring the association of homelessness with the unsuccessful treatment outcomes is presented in a separate table. Table 2 can retain the the title and the contents presenting the association of the sociodemographic and clinical covariates with unsuccessful treatment among the sheltered and homeless population separately, the second objective.

Reviewer #3: Spelling errors in the manuscript may kindly be taken care.

**Summary and General Comments**

Reviewer #1: (No Response)

Reviewer #3: No comments.

PLOS authors have the option to publish the peer review history of their article (what does this mean?). If published, this will include your full peer review and any attached files.

Reviewer #1: Yes: Sonali Sarkar

Reviewer #3: No

Figure Files:

Data Requirements:

Reproducibility:

References

---

## [Editor Report · Decision Letter 4]

26 Sep 2023

Dear Dr. Rodrigues,

We are pleased to inform you that your manuscript 'Factors associated with unsuccessful tuberculosis treatment among homeless persons in Brazil: a retrospective cohort study from 2015 to 2020' has been provisionally accepted for publication in PLOS Neglected Tropical Diseases.

Best regards,

Subash Babu

Academic Editor

Ana LTO Nascimento

Section Editor

---

## [Editor Report · Acceptance letter]

16 Oct 2023

Dear Mr. Silva Rodrigues,

We are delighted to inform you that your manuscript, "Factors associated with unsuccessful tuberculosis treatment among homeless persons in Brazil: a retrospective cohort study from 2015 to 2020," has been formally accepted for publication in PLOS Neglected Tropical Diseases.

Best regards,

Shaden Kamhawi

co-Editor-in-Chief

Paul Brindley

co-Editor-in-Chief
